# How to Deal with Second Line Dilemma in Metastatic Colorectal Cancer? A Systematic Review and Meta-Analysis

**DOI:** 10.3390/cancers11081189

**Published:** 2019-08-15

**Authors:** Antonio Galvano, Lorena Incorvaia, Giuseppe Badalamenti, Sergio Rizzo, Aurelia Guarini, Stefania Cusenza, Luisa Castellana, Nadia Barraco, Valentina Calò, Sofia Cutaia, Giuseppe Currò, Nicola Silvestris, Giordano Domenico Beretta, Viviana Bazan, Antonio Russo

**Affiliations:** 1Medical Oncology, Department of Surgical, Oncological and Stomatological Sciences, University of Palermo, 90127 Palermo, Italy; 2Medical oncology Unit–IRCCS Istituto Tumori “Giovanni Paolo II”, 70124 Bari, Italy; 3Department of Biomedical Sciences and Human Oncology, University of Bari Aldo Moro, 70124 Bari, Italy; 4Medical Oncology Unit Humanitas Gavazzeni, 24125 Bergamo, Italy; 5Department of Biomedicine, Neuroscience and Advanced Diagnostics-BIND, University of Palermo, 90127 Palermo, Italy

**Keywords:** meta-analysis, colorectal cancer, second line, targeted agents, sequence, VEGF, EGFR

## Abstract

Monoclonal antibodies targeting epidermal growth factor receptor (EGFR) or vascular endothelial growth factor (VEGF) have demonstrated efficacy with chemotherapy (CT) as second line treatment for metastatic colorectal cancer (mCRC). The right sequence of the treatments in all RAS (KRAS/NRAS) wild type (wt) patients has not precisely defined. We evaluated the impact of aforementioned targeted therapies in second line setting, analyzing efficacy and safety data from phase III clinical trials. We performed both direct and indirect comparisons between anti-EGFR and anti-VEGF. Outcomes included disease control rate (DCR), objective response rate (ORR), progression-free survival (PFS), overall survival (OS) and G3-G5 toxicities. Our results showed significantly improved OS (HR 0.83, 95% CI 0.72–0.94) and DCR (HR 1.27, 95% CI 1.04–1.54) favouring anti-VEGF combinations in overall population; no statistically significant differences in all RAS wt patients was observed (HR 0.87, 95% CI 0.70–1.09). Anti-EGFR combinations significantly increased ORR in all patients (RR 0.54, 95% CI 0.31–0.96), showing a trend also in all RAS wt patients (RR 0.63, 95% CI 0.48–0.83). No significant difference in PFS and DCR all RAS was registered. Our results provided for the first time a strong rationale to manage both targeted agents in second line setting.

## 1. Introduction

Colorectal cancer (CRC) is the third cause of cancer-related death worldwide. Approximately 25% of patients have metastatic disease at diagnosis and almost 50% of patients with CRC will develop metastases [1]. Of these, less than 10% is suitable for potentially curative resection, whereas the other metastatic colorectal cancer (mCRC) patients are candidates for chemotherapy treatment. However, over the last 15 years, new drugs and their associations, both cytotoxic (fluoropyrimidines, oxaliplatin, irinotecan and, most recently, trifluridine/tipiracil (TAS-102)) and biologic agents (monoclonal antibodies targeting the vascular endothelial growth factor—anti-VEGF: bevacizumab, aflibercept, ramucirumab and regorafenib or the epidermal growth factor receptor—anti-EGFR: cetuximab, panitumumab), have significantly improved objective response rate (ORR), progression free survival (PFS) and overall survival (OS) in mCRC patients [2,3]. The median duration of survival of mCRC patients has increased from 12 months to nearly 36 months due to the availability of several therapeutic options and the success of combining chemotherapy and biological therapies [4]. However, five-year survival following diagnosis of mCRC is still about 14%. In addition, there is no consensus as regards the best therapeutic sequence, primarily in RAS wild-type (wt) mCRC patients. It has been shown that the best survival is obtained when all drugs are administered in a context of a continuum of care [5].

Cytotoxic drug pairs with fluoropyrimidines plus oxaliplatin or irinotecan combination provide higher RR, longer PFS and better OS than fluoropyrimidines alone [6,7,8,9]. The role of triplet combination with 5-FU, oxaliplatin and irinotecan (FOLFOXIRI) has been recently investigated [10,11].

A retrospective analysis showed that the exposure to all three cytotoxics (fluoropyrimidines, oxaliplatin and irinotecan), in various sequences, produce longer survival [9]. The chemotherapeutic regimen alternative to the combination administered during first-line therapy should be offered to patients with good performance status and adequate organ function in second-line treatment. VEGF and EGFR inhibitors have been approved in first- and second-line settings [12]. 

We have analyzed and compared the efficacy and safety of treatment using anti-VEGF (bevacizumab, aflibercept and ramucirumab) or anti-EGFR (cetuximab and panitumumab) agents in second-line in RAS wt mCRC patients, through a systematic review of data reported in the literature.

## 2. Results

The search for literature identified a total of 531 records, of which eight duplicates were excluded; 476 other records were excluded because they were meta-analyses, retrospective or phase I/II studies, reviews, letters or did not report human studies. A total of 47 trials were assessed for eligibility and 39 were excluded because no data about the principal outcomes of our meta-analysis were reported. Finally, a total of eight studies met all the inclusion/exclusion criteria and were included in the meta-analysis (Figure 1, Appendix A).

### 2.1. Direct Comparisons

#### 2.1.1. Anti-VEGF + CT vs CT alone

Five RCTs enrolling 3879 patients evaluated Anti-VEGF (bevacizumab, aflibercept or ramucirumab) + CT vs CT in second line mCRC settings. Pooled results showed statistically significant differences in terms of DCR (RR 1.15, 95% CI 1.07–1.23; *p* = 0.0001), PFS (HR 0.73, 95% CI 0.68–0.78; *p* < 0.00001) and OS (HR 0.81, 95% CI 0.75–0.87; *p* < 0.00001) favouring anti-VEGF combinations. Our analysis reported also a significant trend towards ORR for anti-VEGF combinations (RR 1.46, 95% CI 1.00–2.12; *p* = 0.05). Subgroup analysis concerning RAS/BRAF status showed statistical significance of anti-VEGF combinationx in RAS WT or RAS mutated patients both in term of PFS and OS (only a trend for RAS mutated OS, Appendix A). As regards safety endpoints, in our pooled analysis anti-VEGF combinations have been shown to significantly increase drug-related risk of bleeding (RR 2.40, 95% CI 1.11–5.23; *p* = 0.03), arterial hypertension (RR 4.07, 95% CI 1.82–9.09; *p* = 0.0006), neutropenia (RR 1.34, 95% CI 1.07–1.61; *p* = 0.002), venous thromboembolism (RR 1.40, 95% CI 1.02–1.92; *p* = 0.03) and proteinuria (RR 8.48, 95% CI 4.20–17.13; *p*≤ 0.00001). Grade 3–5 serious adverse events (SAEs) risk was associated to anti-VEGF strategy (RR 1.23, 95% CI 1.14–1.33; *p* ≤ 0.00001, Appendix A). As for most common AEs, anti-VEGF addition did affect diarrhea, vomiting, asthenia and neutropenia risk (RR 1.43, 95% CI 1.31–1.56; *p* < 0.00001, Appendix A).

#### 2.1.2. Anti-EGFR + CT vs EGFR Alone

Three randomized phase III controlled trials (RCTs) enrolling a total of 2944 patients investigated the addition of an anti-EGFR agent (cetuximab or panitumumab) in the same mCRC setting (second line). Our pooled results showed a statistically significant anti-EGFR combination benefit in terms of ORR (RR 2.85, 95% CI 2.01–4.06; *p* < 0.00001), DCR (RR 1.20, 95% CI 1.06–1.36; *p* = 0.005) and PFS (HR 0.71, 95% CI 0.64–0.80; *p* < 0.00001) but not for OS (HR 0.98, 95% CI 0.88–1.10; *p* = 0.31), if compared with CT alone (Appendix A). Considering RAS, our analysis confirmed mutated RAS status as a negative predictive factor for anti-EGFR efficacy both in all the above mentioned endpoints. For safety analysis, EGFR drug-related skin toxicities (RR 24.12, 95% CI 13.11–44.36; *p* < 0.00001) and hypomagnesaemia (RR 13.49, 95% CI 3.20–56.81; *p* = 0.0004) were more associated with anti-EGFR combination regimen. Diarrhea (RR 1.77, 95% CI 1.50–2.09; *p* < 0.00001) risk was significantly related to anti-EGFR strategy. We also registered a trend over neutropenia (RR 1.15, 95% CI 1.00–1.32; *p* = 0.05) and asthenia (RR 1.15, 95% CI 0.99–1.35; *p* = 0.07) while no significant difference was observed for vomiting (RR 0.97, 95% CI 0.81–1.15; *p* = 0.38). Grade 3–5 SAEs were mostly related to anti-EGFR strategy (RR 1.40, 95% CI 1.31–1.50; *p* < 0.00001, Appendix A).

### 2.2. Indirect Comparisons

#### Anti-VEGF vs Anti-EGFR

We used the meta-analytic technique to do an indirect comparison between anti-VEGF and anti-EGFR combination strategy pooled results on clinical (DCR, ORR, PFS and OS) and safety endpoints (most common toxicities and SAEs G3–G5). For clinical endpoints in the overall population, we obtained significant differences favoring anti-VEGF combination in OS (HR 0.83, 95% CI 0.72–0.94) and DCR (RR 1.27, 95% CI 1.04–1.54) while anti-EGFR showed superiority in terms of ORR (RR 0.54, 95% CI 0.31–0.96). No statistical difference in PFS was registered. Comparisons in the RAS wild type subgroup showed a greater benefit for anti-VEGF agents in terms of OS (HR 0.87, 95% CI 0.70–1.09) while Anti-EGFR demonstrated benefit over anti-VEGF in ORR (RR 0.63, 95% CI 0.31–0.96), although they did not reach a statistical relevance. As regards most common safety events, anti-VEGF strategies increased the risk for asthenia (RR 1.34, 95% CI 1.03–1.75), with a trend for neutropenia (RR 1.17 95% CI 0.98–1.40) and vomiting (RR 1.37, 95% CI 0.94–2.00). No difference in terms of diarrhea. (Figure 2; Figure 3; Table 1).

### 2.3. Risk of Bias Assessment

Publication bias tests are necessary in meta-analyses including at least three studies. In our analysis, Egger’s test was calculated including eight trials comparing targeted (anti-VEGF and anti-EGFR) + CT combinations vs CT alone comparisons (anti-VEGF – five trials, anti-EGFR – three trials) showing no statistical significance (*p* = 0.50, Figure 4). The overall quality assessment was evaluated according to the CONSORT checklist statement. We report a good quality of all trials (Figure 5, Appendix A
Appendix A) included in our analysis.

## 3. Discussion

Monoclonal antibodies targeting epidermal growth factor receptor (EGFR) or vascular endothelial growth factor (VEGF) have demonstrated efficacy in combination with chemotherapy as second line for metastatic colorectal cancer (mCRC). However, there is still a paucity of evidence or guidelines suggesting the right sequential treatment in all RAS (KRAS/NRAS) wild type (wt) mCRC. Therefore, the aim of our meta-analysis is to provide a strong evidence on the addition of an anti-EGFR in the second line treatment of mCRC. The results of our research suggest the use of second-line anti-VEGF therapy as a valid and well tolerated option to anti-EGFR therapy in all RAS-WT patients with metastatic mCRC on the basis of a better trend of DCR and OS benefit in overall population, while a better ORR has been showed by using anti EGFR agents. 

### 3.1. Anti-EGFR Combinations and ORR

It has been demonstrated that the use of anti EGFR could only increase the ORR. The SPIRITT trial by Hecht et al., showed that anti-EGFR response are “deeper” (depth of response–DpR) and “earlier” (early tumor shrinkage—ETS—as defined by a reduction >20% of target lesions within 8 weeks) than those seen with anti-VEGF treatments [13,14]. Moreover, Modest et al. have published an update of their previous FIRE-3 trial; the intention to-treat (ITT) population, excluding patients who had died, had experienced 2nd line treatment, secondary resection or re-introduction of the first line treatment. The FIRE 3 study update also reported data on ETS and DpR that confirmed the superiority of anti-EGFR in term of ORR: ETS with FOLFIRI-cetuximab was 68.2% versus 49.1% of the FOLFIRI-bevacizumab arm. DpR also favored the anti-EGFR group with a median value of 48.9% versus 32.3 % of the anti-VEGF group. However, among patients experiencing an ETS, those treated with cetuximab had longer overall survival as compared to those treated with bevacizumab [15]. Also, the CALGB/SWOG 80405 study showed that the ORR results favored first-line anti-EGFR therapy, no significant difference in PFS or OS was detected between the two treatment approaches (chemotherapy plus anti-VEGF vs chemotherapy plus anti-EGFR). As the FIRE-3 trial demonstrated, the first line treatment and specifically the sequence of drug used is more important than the exposure to single agents: the major gain in OS is probably given by the choice of the first line chemotherapy combination. The explanation to this crucial point is maybe given by the pharmacological properties of these agents. Deranger et al. demonstrated, in vitro, in 25 patients specifically progressing to bevacizumab, that a first line treatment with bevacizumab can cause increased levels of VEGF-A that may be responsible for resistance to anti-EGFR therapy [16]. Also, Zeng et al. [17] showed that bevacizumab, by inducing hypoxia, could determine a marked increase in KRAS activity (the GTP-bound KRAS form) and result in an anti-EGFR resistance. On the contrary, anti-EGFR agents could improve the expression of VEGF protein, confirming our hypotheses of using this sequence of drugs [17]. The hypoxia induced by an anti-VEGF may lead to molecular and biological changes in cancer cells, thus cells may acquire resistance by living in a more stressful environment. One of the most important change, described by Zhou et al. is represented by the modifications of the epithelial–mesenchymal transition (EMT), a cellular program characterized by loss of epithelial markers like the deregulation or reduced expression of E-cadherin and the acquirement of a mesenchymal phenotype. In lung cancer cells lines the modified EMT conferred resistance to anti-EGFR agents [18].

### 3.2. Role of Anti-EGFR Combinations in Second-Line

Also, the use of second line anti-EGFR agents can be useful for “secondary resections”, i.e., metastases resection after second line chemotherapy. A retrospective analyses by Adam et al. showed that resection following second-line pre-operative chemotherapy, could bring similar OS compared to what observed after first-line. A larger percentage of patients that underwent a secondary resection received irinotecan based regimen containing anti EGFR agents. This retrospective analysis presented obviously many limitations: chemotherapy regimens were decided by the physician and the algorithms were non standardized, moreover all the patients that did not undergo resection were not evaluated in LiverMetSurvey [19].

The results of the use of second line anti-VEGF agents but mainly the optimal sequence of targeted therapies, has been reported in many different trials; Peeters et al., evaluated the optimal sequence of targeted therapies (EGFR and VEGF inhibitors) in patients with RAS wt mCRC. Patients from the studies PEAK, PRIME and Study 181 were included in the analysis and the results suggested a positive trend towards improved OS for first line panitumumab plus chemotherapy followed by second-line anti VEGF [19]. Also, the EPIC trial [20] and the PICCOLO trial [21] showed no OS benefit had been observed in second line anti EGFR combination treatments. Our results are in line with all the abovementioned studies, providing robust data to define the all RAS WT mCRC approach in second line setting. Our analysis also registered no significant difference in the toxicity profile, only a significant asthenia difference was noted in the anti VEGF combination treatment.

In conclusion, the indirect comparisons of our research showed, in overall mCRC population, a significantly improved OS and DCR for second line anti-VEGF combinations, while no PFS differences were registered. Our anti-VEGF OS trend in all RAS WT subgroup suggest an anti-VEGF strategy beyond first-line progression (bevacizumab or aflibercept-based), reserving an anti-EGFR strategy in those cases who could benefit from secondary resection, according to the anti-EGFR ORR benefit. Our indirect comparison presents some limitations due above all to the different backbone chemotherapy regimens (FOLFOX/4 or 6, FOLFIRI, CAPOX) used in combination with the targeted agent or to the lack of biological agent benefit in one first line setting trial (Giantonio et al. [20]). Also, first-line different proportion of patients according to different strategy (anti-VEGF or anti-EGFR) could suggest a weak OS performance for anti-EGFR combinations because of the small cohort of patient underwent to first-line bevacizumab strategy (Sobrero et al. [21] 13 % beva; Peeters et al. [22] 18% beva; Seymour et al. [6,23] 2% beva), according to data available. Furthermore, other studies did not report efficacy data on BRAF wt cohort; in this case we include the KRAS-NRAS wt population as mentioned in Materials and Methods. Moreover, the other studies should evaluate the role of predictive biomarkers [22,23].

## 4. Materials and Methods 

We searched for results of RCTs comparing targeted therapies (anti-VEGF or anti-EGFR) associated with backbone chemotherapy (CT) regimen (FOLFOX/4 or 6, FOLFIRI, CAPOX) versus CT alone in patients with histologically proven diagnosis of advanced colorectal cancer in progression after first-line chemotherapy. Data available up to March 2019 on Medline (PubMed), EMBASE databases and Cochrane-Library were collected, without language restrictions; relevant abstracts published on the American Society of Clinical Oncology (ASCO) and the European Society of Medical Oncology (ESMO) databases, as well as unpublished data or results from ongoing studies available on the National Institute of Health (NIH) website were also considered as a source of grey literature. We used the following free search terms: “Target Therapy”, “Anti-EGFR”, “Anti-VEGF”, “Bevacizumab”, “Cetuximab”, “Panitumumab”, “Aflibercept”, “Ramucirumab”, “Second line”, “Beyond first”, “Advanced colorectal cancer”, “Metastatic colorectal cancer” (see Appendix A). The outcomes were objective response rate (ORR), disease control rate (DCR) according to Response Evaluation Criteria in Solid Tumors (RECIST) ver 1.0 or 1.1, progression-free survival (PFS) and overall survival (OS). Safety data (AEs–adverse events and SAEs–serious adverse events) were included in our analysis according to Common Terminology Criteria for Adverse Events (CTCAE) grading. We excluded: 1) not randomized, retrospective and crossover studies; 2) studies assessing the association of more anti-EGFR or anti-VEGF agents, as well as the combination of drugs from both classes. We also excluded trials in which data were unavailable, ongoing studies and studies with small sample size (less than 10 patients for arm). To minimize the risk of bias, we excluded observational trials. In case of articles with follow-up over time, we decided to include the most updated and methodically valid. Two review authors (A.G. and A.G.) independently screened articles for inclusion and were responsible for data extraction and assessment. Finally, a total of eight articles (five anti-VEGF and three anti-EGFR) were included in our study [20,21,22,23,24,25,26,27]. Details about studies structure, participants, rules, efficacy and safety were recorded. Incongruences and disagreements were solved by discussing with another author (A.R.). We made a quality analysis of selected trials following the criteria reported in the Cochrane Handbook for Systematic Reviews [28] of Interventions including: allocation concealment; blinding of participants, personnel and outcome assessors; incomplete outcome data; sequence generation; elective outcome reporting; other sources of bias. For each study we defined “Yes” as at low risk of bias and as “No” at high risk of bias. We define also “unclear” if there were insufficient data for a precise judgement. The risk of selective outcome reporting bias was also evaluated by two independent reviewers (A.G. and S.C.) and disagreement were solved by consensus.

The analyzed outcomes for both direct and indirect comparisons were OS, PFS, DCR, ORR and grade 3–5 (G3–G5) most common AEs. In addition, ORR, DCR, PFS and OS were stratified, when possible, according to the tumor RAS (K- and N-RAS) and BRAF expression status. We performed also a pre-planned subgroup analysis on the basis of RAS/BRAF status. We used hazard ratios (HRs) to assess the association for PFS and OS, with a 95% confidence interval (CI). For all other outcomes (ORR, DCR and G3–G5 AEs) we calculated the total number of events over the total patient number randomized in each group, thus using the Risk Ratios (RRs) as the measure of association. In the first phase of the study, using meta-analysis techniques, we performed a direct comparison including all the RCTs evaluating the efficacy of a targeted therapy (anti-VEGF or anti-EGFR) associated with standard CT versus standard CT alone, by calculating the logarithm of the HR (logHRs) or RR (logRRs) and its relative standard error (SE) for all the RCTs included in this phase. Afterwards, we calculated pooled data of every comparison. 

We took final estimates from previous meta-analyses in order to obtain pooled estimates of HR and RR needed for the indirect comparison [29]. The method used in the indirect comparison was the one described by Bucher and Glenny, extended to calculate the HR. We chose this method for its ability to maintain the randomization advantage of each trial providing an estimate of the comparison between treatments [30,31]. Assuming that VEGF_ST_ is the estimate of the direct comparison between anti-VEGF and chemotherapy and EGFR_ST_ is the estimate of the direct comparison between anti-EGFR and chemotherapy, then the estimate of the indirect comparison between anti-VEGF and anti-EGFR can be calculated as follows: VEGF/EGFR_indirect(logHR or logRR) = VEGF_ST_(logHR or logRR) - EGFR_ST_(logHR or logRR). The variance can be obtained with the following computation: Var(logVEGF/EGFR_indirect) = Var(logVEGF_ST_) + Var(logEGFR_ST_) [31]. 

Heterogeneity between studies was evaluated using I-squared test. If I-squared value was higher than 50%, with a high risk of heterogeneity, we performed the meta-analysis using the random effect-based model by Der Simonian and Laird; for I-squared lower than 50% we used the fixed-effect based Mantel-Haenszel model [28,32]. In our indirect meta-analysis PFS and OS with HR < 1 suggest a better efficacy of anti-VEGF agents, whereas ORR, DCR and AEs with RR < 1, suggest a better efficacy of anti-EGFR agents. As regards the risk of bias across studies, we performed a publication bias analysis using Egger’s test providing the respective Funnel Plot. The manuscript was done and reported according to the PRISMA-guidelines for reporting on systematic reviews [33]. The meta-analysis was performed using the Cochrane RevMan ver. 5.3 software and *p*-values were considered statistically significant if *p* < 0.05.

## 5. Conclusions

To our knowledge, our meta-analysis results support, for the first time, a trend towards improved OS and DCR for anti-VEGF combinations in second line mCRC and thus providing to clinicians a robust and encouraging scientific evidence to select the best strategy for every patient according to mutational status, clinical conditions and toxicities. Moreover, this scenario should change with the improvement of immunotherapy. Nonetheless, prospective phase III studies are needed to evaluate the optimal treatment sequencing of biological therapies.

## Figures and Tables

**Figure 1 cancers-11-01189-f001:**
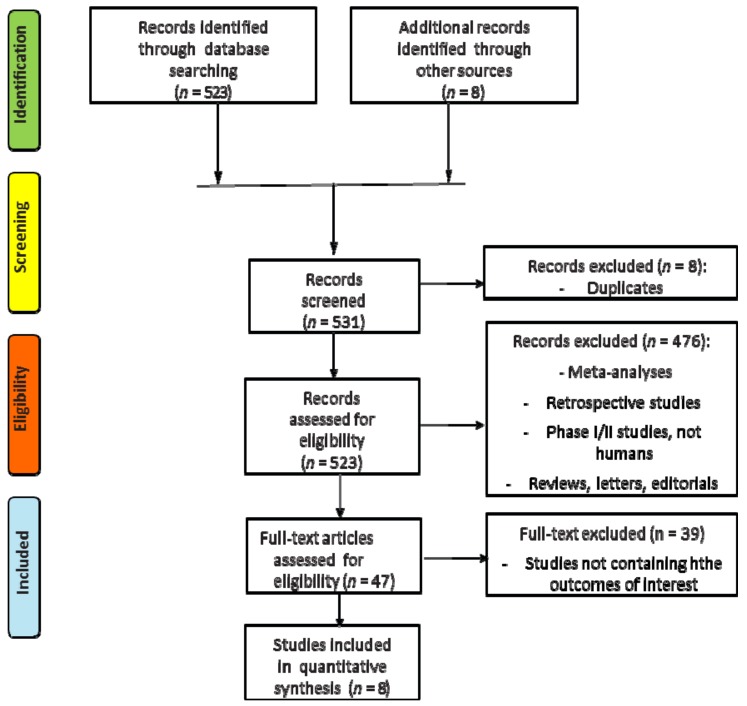
Flow diagram (CONSORT) for the meta-analysis included studies (according to the PRISMA statement).

**Figure 2 cancers-11-01189-f002:**
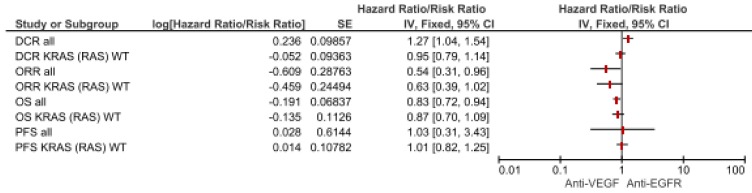
Forest plot of anti-VEGF vs anti-EGFR combination therapy for clinical endpoints according to mutational status. Abbreviations: disease control rate (DCR); overall response rate (ORR); progression-free survival (PFS); overall survival (OS).

**Figure 3 cancers-11-01189-f003:**
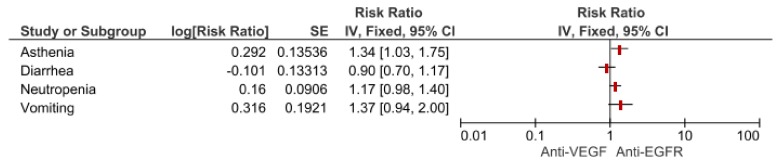
Forest plot of anti-VEGF vs anti-EGFR combination therapy for most common toxicities.

**Figure 4 cancers-11-01189-f004:**
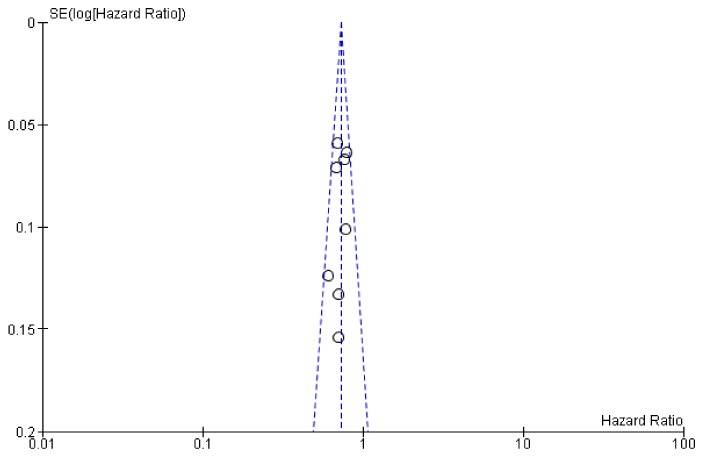
Plot for publication bias assessment (Egger’s test *p* > 0.05).

**Figure 5 cancers-11-01189-f005:**
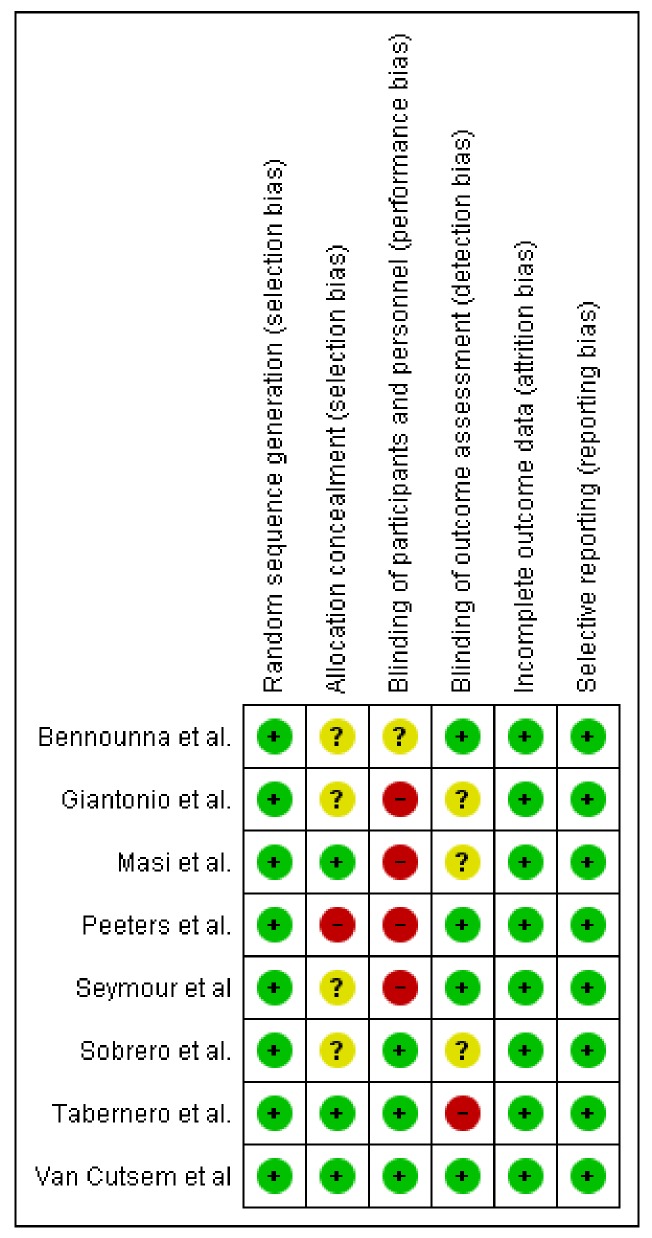
Bias summary: Review authors’ judgements about each risk of bias item for each included study.

**Table 1 cancers-11-01189-t001:** Results.

	Anti-VEGF vs Anti-EGFR							
**Cohort**	ORR RR (95% CI)	DCR RR (95% CI)	PFS HR (95% CI)	OS HR (95% CI)	Asthenia RR (95% CI)	Diarrhea RR (95% CI)	Neutropenia RR (95% CI)	Vomiting RR (95% CI)
Overall population	0.54 (0.31–0.96)	1.27 (1.04–1.54)	1.03 (0.31–3.43)	0.83 (0.72-0.94)	1.34 (1.03–0.94)	0.90 (0.70–1.17)	1.17 (0.98–1.40)	1.37 (0.94–2.00)
K-NRAS/BRAF Wild type	0.63 (0.39–1.02)	0.95 (0.79–1.14)	1.01 (0.82–1.25)	0.87 (0.82–1.25)	NA	NA	NA	NA

ORR: Overall Response Rate; DCR: Disease Control Rate; PFS: Progression-free Survival; OS: Overall Survival; CI: confidence intervals; RR: Risk ratio; HR: Hazard Ratio; NA: Not Applicable.

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
