# Peer review of "How to Deal with Second Line Dilemma in Metastatic Colorectal Cancer? A Systematic Review and Meta-Analysis"

_cancers, 2019, doi:10.3390/cancers11081189_

Round 1
Reviewer 1 Report
Galvano et al present a meta analyse of phase III randomized controlled trials investigating the role of anti-EGFR or anti-VEGF therapies as second line biologics in treating metastatic colorectal cancer. The authors report improved overall survival in patients treated with anti-VEGF as a second line agent, however second line therapy with anti-EGFR elicited better disease control and overall response rates. When a subgroup analyses of RAS wt patients was performed there was a non-significant trend (p value?) towards an increased overall response rate compared with anti-VEGF therapy. These findings must be interpreted with case and have several limitations (as outlined in the authors discussion). Nevertheless the findings are important, and may inform the design of future clinical trials to validate the presented findings. I have but a few minor points.
Specific comments
1) In figure two, can the authors confirm that the groups that are not marked WT are all patients (or alternatively are RAS mutant?)
2) P values for comparisons should be provided throughout.
3) The discussion is disorganized in places and is hard to follow. This could be improved by segmenting the discussion into different parts.
Author Response
Specific comments
In figure two, can the authors confirm that the groups that are not marked WT are all patients (or alternatively are RAS mutant?)
R: Thank you for your suggestion. We modified the figure 2 (all vs WT)
P values for comparisons should be provided throughout.
Thank you. P values have been added fo all figures but those for indirect comparisons because they are not available
The discussion is disorganized in places and is hard to follow. This could be improved by segmenting the discussion into different parts.
R: Thank you for your comment that could improve the clinical meaning of our research. We divided the discussion section into some different paragraph in order to simplify the reading, as you suggested.
Reviewer 2 Report
This manuscript is a meta-analysis of phase 3 clinical trials in metastatic colorectal cancer, focusing on efficacy of 2nd-line targeted therapies targeting EGFR or VEGFR. In particular, they focus on the order in which the drugs should be given (in the abstract they mention "our results provide ... a strong rationale to manage both targeted agents in the second line setting.")
The main analysis is a pooled forest plot approach comparing (Anti-VEGF+CT vs CT) to (Anti-EGFR+CT vs CT) and the authors conclude that Anti-VEGF showed OS and DCR response measures overall. However, this study does not provide "robust evidence" as the authors claim as to the 'sequence' in which the drugs should be given.
I am not sure how the authors pooled results or how they interpreted it. It looks clear to me on the forest plot that HR/RR >1 favors anti-EGFR and <1 favors anti-VEGF. Most of the authors' conclusions are in line with this, but they also say that DCR HR=1.27 favors anti-VEGF and ORR RR=0.54 favors anti-EGFR (line 28, line 113) which is very confusing to me.
I am wondering if this is all errors in interpretation or in writing. Even as a native English speaker and reader, this review was a little difficult to understand, mainly due to grammatical issues. There are so many, I list only a few below:
- line 38, 'third cause of cancer-related death' .. i think you are missing the word 'highest'
- line 24, missing spaces
- line 45, line 64, 'e', probably means 'and'?
- objective response rate is defined as RR in line 45 but both ORR and RR are used throughout the paper, and RR is also defined as 'risk ratio' line 131.
- RCT is defined line 215 but used starting line 94.
- FP line 58 not defined, assume the authors mean fluoropyrimidines.
- Fig 2. what is SE? standard error?
- line 203: "patients undergone to an anti-VEGF regimen." do you mean "patients who underwent an anti-VEGF regimen?"
There are many sentences that I can't understand, even if I try to guess the meaning:
- line 84, "trend over RAS mutated OS." what does this mean?
- line 59, 'hesitates in a longer survival." what does this mean?
- line 137, "an average good quality." what does this mean?
- line 166, "However, any significant difference in PFS or OS was detected." what does this mean?
Materials and methods are the 4th section, between DIscussion and Conclusion, but are better suited earlier on in the paper. The flow is disrupted.
The third table in Table S1 should all be BRAF? The last two columns may be mislabelled as KRAS.
Can the authors discuss why Anti-VEGFR showed much stronger ORR than Anti-EGFR but weaker DCR? I think they touched on it tangentially in line 156, but could elaborate.
There are some interesting discussion points, but it is hard to read in one huge paragraph. The point about first-line treatment affecting the result of the second-line is buried in there. If the data is available, the authors should look into the distribution of first line treatments patient underwent in each trial, and discuss how this may have affected their second line result.
The overall message is confusing. The authors suggest using anti-VEGF as the second line therapy of choice in the conclusion, but in the discussion seems to be the opposite: 1) there is a lot of text about how anti-VEGF could could increase resistance to anti-EGFR therapy, and 2) "on the contrary, anti-EGFR agents could improve expression of VEGF, confirming our hypothesis of using this sequence of drugs)" which suggests to the reader that anti-VEGF could work well after anti-EGFR. The abstract doesn't say which combination is better, but does say they provide "a strong rationale to manage both targeted agents."
Author Response
I am not sure how the authors pooled results or how they interpreted it. It looks clear to me on the forest plot that HR/RR >1 favors anti-EGFR and <1 favors anti-VEGF. Most of the authors' conclusions are in line with this, but they also say that DCR HR=1.27 favors anti-VEGF and ORR RR=0.54 favors anti-EGFR (line 28, line 113) which is very confusing to me.
R: Thank you. We really appreciate your comments because they will help us make the manuscript more understandable to the readers. Our main results (figure 2) showed an OS and DCR all cohort advantage for anti VEGF combination therapy (HR: 0.83; 0.72 – 0.94 and HR: 1.27; 1.04 – 1.54 respectively). An OS significant trend was seen for the RAS wild type population too. So, patients affected by RAS WT mCRC and undergone to an anti-VEGF first line combination therapy could benefit from a second line anti-VEGF combination therapy rather than an anti-EGFR combinatory regimen, as suggested by our results. As regards the figure 2 interpretation, as we declared in materials and methods section, survival outcomes (PFS and OS) have been evaluated using hazard ratios while proportional outcomes (ORR and DCR) have been evaluated using Risk Ratios. Consequently, DCR RR equal to 1.27 means that anti VEGF have a DCR probabibility > than anti EGFR. At the same time, ORR RR equal to 0.54 means that anti VEGF have an ORR probabibility < than anti EGFR.
I am wondering if this is all errors in interpretation or in writing. Even as a native English speaker and reader, this review was a little difficult to understand, mainly due to grammatical issues. There are so many, I list only a few below:
- line 38, 'third cause of cancer-related death' .. i think you are missing the word 'highest'
R: Thank you. We described colorectal cancer as the third cause of cancer-related death according to the [1] reference.
- line 24, missing spaces
R: We corrected the text as you suggested
- line 45, line 64, 'e', probably means 'and'?
R: We corrected the text as you suggested
- objective response rate is defined as RR in line 45 but both ORR and RR are used throughout the paper, and RR is also defined as 'risk ratio' line 131.
R: Thank you. We corrected the text using “RR” as “Risk ratio” and “ORR” as “Objective response rate”
- RCT is defined line 215 but used starting line 94.
R: We corrected the text as you suggested
- FP line 58 not defined, assume the authors mean fluoropyrimidines.
R: We corrected the text as you suggested
- Fig 2. what is SE? standard error?
R: Thank you. Yes it is (see line 261)
- line 203: "patients undergone to an anti-VEGF regimen." do you mean "patients who underwent an anti-VEGF regimen?"
R: Thank you. We corrected the text as you suggested
There are many sentences that I can't understand, even if I try to guess the meaning:
- line 84, "trend over RAS mutated OS." what does this mean?
R: Thank you. We corrected the text. We mean that for OS our results did not reach a statistically significant value
- line 59, 'hesitates in a longer survival." what does this mean?
R: Thank you. We corrected the text. We mean that drug exposure could lead to longer survival.
- line 137, "an average good quality." what does this mean?
Thank you. We mean the our systematic research produce good quality papers
- line 166, "However, any significant difference in PFS or OS was detected." what does this mean?
Thank you. We modify the text as you suggested.
Materials and methods are the 4th section, between DIscussion and Conclusion, but are better suited earlier on in the paper. The flow is disrupted.
R: Thank you for you suggestions. We already followed the journal author’s guidelines as below specified:
General Considerations
Research manuscripts should comprise:
Front matter: Title, Author list, Affiliations, Abstract, Keywords
Research manuscript sections: Introduction, Results, Discussion, Materials and Methods, Conclusions (optional).
The third table in Table S1 should all be BRAF? The last two columns may be mislabelled as KRAS.
R: Thank you. We modified the table to let reader to better understand the key message
Can the authors discuss why Anti-VEGFR showed much stronger ORR than Anti-EGFR but weaker DCR? I think they touched on it tangentially in line 156, but could elaborate.
R: Thank you for your comment. As we already above specify, anti-VEGF ORR<1 means that antiVEGFs have a lower chance to reach a response if compared to anti-EGFRs. On the contrary, anti-VEGFs DCR>1 meas that antiVEGFs have a higher chance to obtain a disease control respect to anti-EGFRs strategy. So, we already discuss the role of ORR for anti-EGFRs combinations in the discussion section strarting from line 158.
There are some interesting discussion points, but it is hard to read in one huge paragraph. The point about first-line treatment affecting the result of the second-line is buried in there. If the data is available, the authors should look into the distribution of first line treatments patient underwent in each trial, and discuss how this may have affected their second line result.
R: Thank you for your comment that could improve the clinical meaning of our research. We divided the discussion section into some different paragraph in order to simplify the reading, as you suggested. Furthermore, we discuss the first-line proportion of patients affecting final OS, according to antiVEGF or anti-EGFR different strategy in the discussion section.
The overall message is confusing. The authors suggest using anti-VEGF as the second line therapy of choice in the conclusion, but in the discussion seems to be the opposite: 1) there is a lot of text about how anti-VEGF could could increase resistance to anti-EGFR therapy, and 2) "on the contrary, anti-EGFR agents could improve expression of VEGF, confirming our hypothesis of using this sequence of drugs)" which suggests to the reader that anti-VEGF could work well after anti-EGFR. The abstract doesn't say which combination is better, but does say they provide "a strong rationale to manage both targeted agents."
R: Thank you for your comment. We will try to better explain why we retain our discussion in line with conclusions. As we specify in discussion section, our meta-analysis focuses management of RAS wt mCRC underwent to an anti-VEGF combination regimen in first line setting. Our aim is to suggest which is the best second line option in this setting (anti-VEGF or anti-EGFR). Then, there are two different options after first line progression disease: anti VEGF beyond progression /Aflibercept or anti-EGFR regimens. So:
Regards “lot of text about how anti-VEGF could increase resistance to anti-EGFR therapy” we report this data to discuss how anti-VEGF first-line could negatively affect the efficacy of anti-EGFR second line, suggesting a role for anti-VEGF in second line (beyond progression or Aflibercept).
Regards "on the contrary, anti-EGFR agents could improve expression of VEGF, confirming our hypothesis of using this sequence of drugs", we intend that second line anti-VEGF is molecularly supported by data after anti-EGFR regimens.
Regards "a strong rationale to manage both targeted agents” stated in the abstract, we mean that both targeted agents could have a role in RAS wt second line mCRC: anti-VEGF is our metanalysis best choice for the largest proportion of patients which clinical aim is the disease control, while anti-EGFR is to suggest when the clinical aim is to obtain a high probability of disease response rather than disease control.
Reviewer 3 Report
The present article is a well written manuscript with very important database that will be beneficial for the treatment of metastatic colorectal cancers. Any metastatic cancer is a challenge to treat in the clinics in present day, especially from breast and colorectal origins. Role of EGFR and VEGFR in metastasis is well established and various anti-VGFR therapy are there in different stages of clinical trials. In this context, the meta-analysis results that have been presented in this manuscript will be considerably helpful for further studies and decision making in the treatments. The data have been well presented and discussed in details. It worth mentioning that the authors not only performed the efficacy comparison analysis, but also presented the comparative hazard analysis. This increases the importance of the manuscript. Though the present article was submitted as a review, the article has sufficient analytical data to be considered as research article. Hence I recommend the present article to be published only after the following minor revision:
1. Many part of figure 1 are not easy to read due to the small font size and strong color tones. Please change it accordingly so that it becomes easy to read. Using larger fonts and changing the color scheme may solve the issue.
2. In figure 4, two words in the legend/description appear to be connected, please split them.
3. Legend/description is incomplete. Please correct.
Author Response
. Many part of figure 1 are not easy to read due to the small font size and strong color tones. Please change it accordingly so that it becomes easy to read. Using larger fonts and changing the color scheme may solve the issue.
R: Thank you. We modify the figure 1 as you suggested.
2. In figure 4, two words in the legend/description appear to be connected, please split them.
R: Thank you. We modified the legend as you suggested
3. Legend/description is incomplete. Please correct.
R: Thank you. We modified the legend as you suggested